# HIERARCHICAL SIDE-TUNING FOR VISION TRANSFORMERS

## ABSTRACT

Fine-tuning pre-trained Vision Transformers (ViT) has consistently demonstrated promising performance in the realm of visual recognition. However, adapting large pre-trained models to various tasks poses a significant challenge. This challenge arises from the need for each model to undergo an independent and comprehensive fine-tuning process, leading to substantial computational and memory demands. While recent advancements in Parameter-efficient Transfer Learning (PETL) have demonstrated their ability to achieve superior performance compared to full fine-tuning with a smaller subset of parameter updates, they tend to overlook dense prediction tasks such as object detection and segmentation. In this paper, we introduce Hierarchical Side-Tuning (HST), a novel PETL approach that enables ViT transfer to various downstream tasks effectively. Diverging from existing methods that exclusively fine-tune parameters within input spaces or certain modules connected to the backbone, we tune a lightweight and hierarchical side network (HSN) that leverages intermediate activations extracted from the backbone and generates multi-scale features to make predictions. To validate HST, we conducted extensive experiments encompassing diverse visual tasks, including classification, object detection, instance segmentation, and semantic segmentation. Notably, our method achieves state-of-the-art average Top-1 accuracy of **76.0%** on VTAB-1k, all while fine-tuning a mere **0.78M** parameters. When applied to object detection tasks on COCO testdev benchmark, HST even surpasses full fine-tuning and obtains better performance with **49.7** box AP and **43.2** mask AP using Cascade Mask R-CNN.

## 1 INTRODUCTION

Recently, large Vision Transformers (ViTs) have achieved remarkable success across various visual tasks (Dosovitskiy et al.; Liu et al., 2021; He et al., 2022a; Radford et al., 2021). Inspired by the success of large language models like (Brown et al., 2020; Devlin et al., 2018), there is a growing enthusiasm for harnessing the pre-trained knowledge embedded within ViTs to elevate the performance in downstream tasks. However, the rapid increase in model size has made direct fine-tuning of these pre-trained models for downstream tasks impractical due to the associated storage overhead. To address this challenge, many studies have introduced Parameter-efficient transfer learning (PETL) (Lian et al., 2022; Hu et al., 2021; Jia et al., 2022; Houlsby et al., 2019; Sung et al., 2022) to develop a high-performing system without the necessity of training an entirely new model for each task. PETL methods operate by selecting a subset of pre-trained parameters or introducing a limited number of trainable parameters into the backbone, while keeping the majority of the original parameters fixed. In the field of computer vision, PETL methodologies can be broadly categorized into two primary groups: adapters and prompt tuning. Adapters involve the incorporation of compact modules into transformer blocks, while prompt tuning consists of concatenating small parameters with input embeddings.

Despite the significant success achieved by these PETL methods, it is important to highlight that most of these techniques have primarily been designed for recognition tasks. When they are extended to accommodate dense prediction tasks such as object detection and segmentation, they still have a large gap compared to the full fine-tuning, which might be due to the fact that dense prediction tasks are fundamentally different from classification tasks. To address this performance gap, we introduce a more versatile PETL method known as Hierarchical Side-Tuning (HST). As shown in

Figure. 1, we depart from existing methods by segregating the majority of trainable parameters from the backbone. This division enables us to establish a Hierarchical Side Network (HSN), capable of producing pyramidal outputs and effectively adapting the entire model to diverse tasks In order to effectively aggregate crucial knowledge from the pre-trained backbone and incorporate it into HSN, we begin by reassessing and reconfiguring the usage of trainable visual prompts, referred to as Meta-Token (MetaT) within our framework. Furthermore, we meticulously craft the Adaptive Feature Bridge (AFB) to bridge and preprocess intermediate activations from the backbone, facilitating a seamless flow of information injection. Within HSN, we propose an innovative Side block as the foundational component of HSN construction. The Side block comprises two primary modules: the Cross-Attention module and the Feed-Forward Neural Network (FFN). It takes intermediate activations and multi-scale features as input, enabling targeted feature fusion based on inputs of varying granularity. Our HSN exhibits the capability to generate multi-scale output features akin to hierarchical ViT variants (Liu et al., 2021; Lin et al., 2023; Wang et al., 2021). By incorporating prior knowledge related to images into the pre-trained backbone, our model is well-equipped for handling demanding dense prediction tasks. Notably, despite the necessity of propagating through two distinct networks during inference, HST does not necessarily increase inference time significantly because computations for the same level of the backbone network and HSN can be performed in parallel.

We conduct comprehensive experiments on HST, including 19 vision recognition tasks of VTAB-1k, object detection, instance segmentation and semantic segmentation. Overall, HST achieves state-of-the-art (SOTA) performance compared to existing PETL methods with comparable trainable parameters. In comparison to full fine-tuning method, HST exhibited a significant performance improvement of 10.4% (76.0% vs. 65.6%) in terms of average Top-1 accuracy on VTAB-1K with merely 0.78M trainable parameters. Furthermore, our HST outperformed other PETL methods by a substantial margin and achieve performance levels closest to full fine-tuning method on COCO (Lin et al., 2014) and ADE20K (Zhou et al., 2017) for dense prediction tasks.

In summary, the contributions of this paper are as follows.

• We introduce a novel parameter-efficient transfer learning approach named Hierarchical Side-Tuning (HST), which separates the majority of trainable parameters from the pre-trained model. This strategic partitioning enables the creation of a Hierarchical Side Network (HSN), notable for its ability to generate multi-scale features optimized specifically for dense prediction tasks.

• We enhance our approach by incorporating Meta-Token (MetaT) with input embeddings in Transformer block, which allows us to acquire additional crucial intermediate features. Then, we developed Adaptive Feature Bridge (AFB) to facilitate a smooth flow of information injection, and Side blocks within HSN to optimize the integration of multi-scale features and intermediate features extracted from pre-trained backbone, resulting in improved efficiency.

• We evaluate our HST on several widely used benchmarks, including image classification, object detection, instance segmentation and semantic segmentation. The experimental results consistently demonstrated that our HST outperformed the existing PETL methods in all these tasks, showcasing its remarkable adaptability.

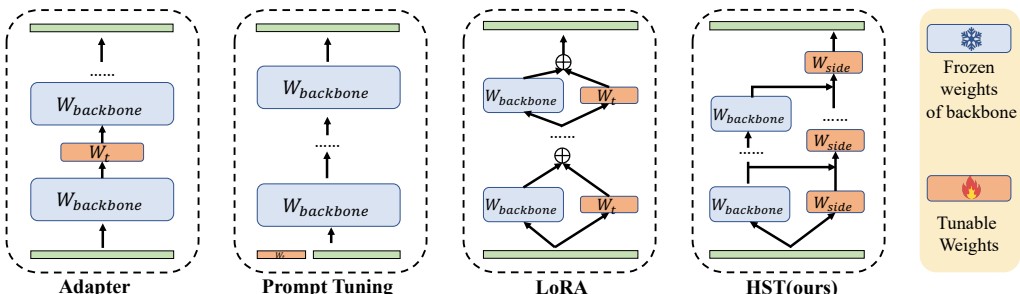

Figure 1: **Previous paradigm *vs.* our paradigm,** including Adapters, Prompt Tuning, LoRA and our Hierarchical Side-Tuning (HST).

## 2 RELATED WORK

**Vision Transformer.** Transformers (Vaswani et al., 2017) have showcased remarkable performance on computer vision tasks. ViT (Dosovitskiy et al.) is the first work to generalize the Transformer to the vision task without much modification. Subsequently, inspired by its vast success, numerous vision Transformer models (Liu et al., 2021; Lin et al., 2023; Touvron et al., 2021; Tu et al., 2022; Dong et al., 2022; Chu et al., 2021; Wang et al., 2021; Chen et al., 2021a) have been proposed following the pioneering work of ViT. The majority of these models progressively grow in size to reach state-of-the-art (SOTA) outcomes and learn the rich representations. In such a situation, the pre-trained Transformer models exhibit significant potential for adaptation to diverse domains. Adopting these pre-trained Transformer models for downstream tasks can alleviate the training difficulty and lead to the swift attainment of promising results. However, addressing the challenge of adapting the pre-trained ViT to downstream tasks in a manner that is both parameter and memory efficient remains a pivotal open issue.

**Decoders for ViT.** ViT is a powerful alternative to standard ConvNets for image classification. However, the original ViT is a plain, non-hierarchical architecture. As a result, it cannot be relatively straightforward to replace a ConvNet with the backbone for dense prediction. Therefore, researchers try to push the frontier of plain backbones for dense prediction. Recently, UViT (Chen et al., 2021b) uses single-scale feature maps for the detector heads, which modifies the architecture during pre-training. Unlike UViT, several studies (Li et al., 2021; 2022) focus on using multi-scale adaptor to maintain the task-agnostic nature of the backbone. Furthermore, SETR (Zheng et al., 2021) develops several CNN decoders for semantic segmentation. Vit-Adapter (Chen et al., 2022b) design a spatial prior module and two feature interaction operations to reorganize multi-scale features for dense prediction, which improving the ViT's weakness of single-scale representation.

**Parameter-Efficient Transfer Learning.** As model sizes continue to expand rapidly, there has been a growing focus on Parameter-Efficient Transfer Learning (PETL) (Liu et al., 2022; Lester et al., 2021; Mao et al., 2021; He et al., 2022b; 2021). PETL targets re-adopting a large-scale pre-trained model as the starting point and only fine-tuning a few lightweight modulesto achieve fair performance competitive to a fully tuned one. Adapter-based and prompt-based tuning stand as two main paradigms for pre-trained models. As depicted in Figure 1, Visual Prompt Tuning (VPT) (Jia et al., 2022) utilizes prompts, comprised of trainable tokens, within the input sequence of the vision Transformer. However, VPT necessitates a search for the optimal prompt length for each specific downstream task, a process that can be time-consuming. Adapter (Houlsby et al., 2019) proposes an MLP-like module with two fully connected layers inserted into the backbone. The adapter presents an effective design, initially reducing the dimensionality of the original features through a single nonlinear layer and subsequently mapping them back to their original dimensions. Unlike injecting trainable modules into the transformer blocks, LoRA (Hu et al., 2021) learns to optimize a low-rank decomposition matrix with a low intrinsic dimension to project the matrices of multi-head self-attention. NOAH (Zhang et al., 2022) propose a prompt search algorithm to automatically combine the adapter, prompt tuning and LoRA. Moreover, SSF (Lian et al., 2022) inserts scale and shift factors into vision models to adapt downstream tasks. Side-Tuning (Zhang et al., 2020a) uses an additive side network, which sums its representation with the backbone network in the last layer. LST (Sung et al., 2022) has similarities to Side-Tuning, but it aims to reduce the memory requirement of current PETL methods. Different from these methods, we design a trainable side network using a pyramid architecture while also taking input priors into account, and use it as ViT's decoder to generate multi-scale output features. This approach enhances its suitability for dense prediction tasks.

## 3 HIERARCHICAL SIDE TUNING

### 3.1 OVERALL ARCHITECTURE

As illustrated in Figure 2, for ViT, the input image is initially passed through the patch embedding, and then the non-overlapping patches are flattened to process through $L$ Transformer encoder layers. Conversely, for HSN, we initiate the process by routing the input image through convolutional stem, introducing local spatial contexts from the input image. The HSN is structured into four stages, each with downsampling rates of $\{4, 8, 16, 32\}$. In this way, we obtain four target resolutuions,

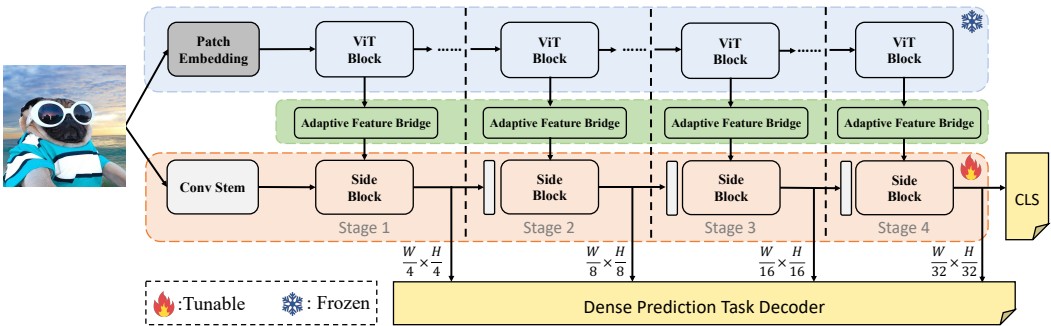

Figure 2: **Overall architecture of HST.** The Blue Section represents the plain ViT, with its weights kept frozen. The Green Section is referred to as the Adaptive Feature Bridge (AFB), which serves the crucial role of bridging and preprocessing intermediate activations derived from the ViT. The Pink Section is the proposed Hierarchical Side Network (HSN), composed of a convolutional stem followed by a sequence of $L$ Side blocks.

forming a feature pyramid akin to those seen in hierarchical networks (He et al., 2016; Liu et al., 2021). It is noteworthy that we align the number of Side blocks with the number of ViT's blocks and evenly distribute them across these four stages, and the overall information flow progresses from the backbone to the hierarchical side network.

## 3.2 META TOKEN

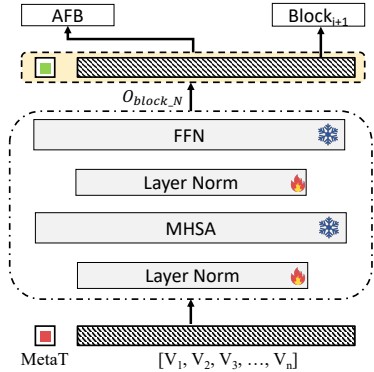

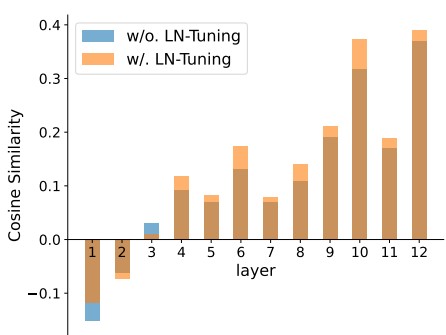

Figure 3: **Meta Token and layer norm tuning.** The MetaT is updated under the effect of attention bias.

Figure 4: Comparisons of cosine similarity between MetaT and input tokens.

Existing prompt-based tuning techniques (Jia et al., 2022; Lester et al., 2021; Li & Liang, 2021) have two significant limitations. First, they rely on manual selection to determine the optimal prompt length for each task, and sometimes the number of prompts can even extend to several hundred, placing a substantial burden on both training and inference. Second, the output features of prompts are discarded after passing through the Transformer layer, resulting in the underutilization of valuable learning information contained within the prompts. To this end, we suggest constraining the number of trainable prompts to a few number (usually N=1), which we refer to as Meta Token (MetaT) ( Figure 3). Furthermore, instead of discarding the output feature of prompts, we input them into Adaptive Feature Brige as intermediate activations together with the output of patch tokens. However, we observe that the output feature distribution of MetaT diverges from that of the patch tokens. This disparity hinders our ability to effectively model within the HSN. To address, we propose the fine-tuning of the layer normalization (LN) layer within the Transformer. Tuning the LN layers can efficiently alter the mean and variance of the feature distribution, thereby aiding in preserving the relative magnitudes among different features within the same sample. Figure 4 illustrates the cosine similarity between the output features of MetaT and the patch tokens in each Transformer layer. It is evident that, through LN tuning, MetaT becomes increasingly more aligned with the vector direction of the patch tokens across layers. This alignment enables us to effectively leverage the output features of learnable MetaT in our tuning framework.

### 3.3 Adaptive Feature Bridge

Considering the mismatch in shapes and dimensions between the intermediate activations derived from ViT and the multi-scale features within the HSN, direct injection becomes unfeasible. Consequently, we introduce a mid-processing module called the Adaptive Feature Bridge (AFB), which comprises two pivotal operations: Dual-Branch Separation and Linear Weight Sharing.

**Dual-Branch Separation.** As shown in Figure 5, the output features of MetaT and patch tokens are initially passed through a linear layer for dimension transformation to ensure alignment with the various stages within the HSN. Subsequently, we divide the processed features into two distinct branches. All patch tokens are globally averaged to yield a single token, known as 'GlobalT', which

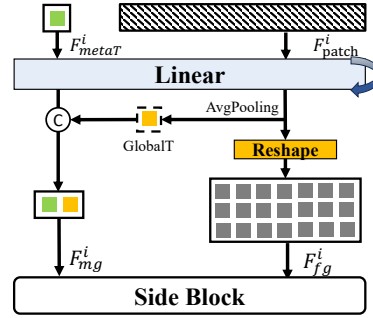

Figure 5: Adaptive Feature Bridge.

is then concatenated with MetaT to form one of the input branches $\mathcal{F}_{mg}^i$. The other branch, $\mathcal{F}_{fg}^i$, utilizes bilinear interpolation to reshape the patch tokens. This reshaping operation aligns the resolution with that of the corresponding stage's feature within the HSN. The whole process can be formulated as follows:

$$\mathcal{F}_{mg}^i = [W_j\mathcal{F}_{MetaT}^i, AvgPooling(W_j\mathcal{F}_{patch}^i)]; \mathcal{F}_{fg}^i = \mathcal{T}(W_j\mathcal{F}_{vit}^i) \tag{1}$$

where $i$ denotes $i$-th ViT block's output, and $W_j$ is the weight matrices of linear layer in $j$-th stage.

**Linear Weight Sharing.** We propose to share the linear weight in AFB for different intermediate features. Specifically, every AFB within the same stage share a common linear layer. This approach offers the distinct advantage of reducing the number of trainable parameters. Simultaneously, it enables information interaction among features within the same stage, thereby achieving effects comparable to those obtained with multiple linear layers.

### 3.4 Side Block

In this section, we detail a novel Side block that forms the fundamental building block of HSN construction. The Side block comprises a cross-attention layer and a feed-forward network (FFN), which collectively empower the modeling of intermediate features from ViT and multi-scale features. Considering the unique characteristics of the two input branches, we introduce them into the Side block through distinct approaches, specifically termed Meta-Global Injection and Fine-Grained Injection.

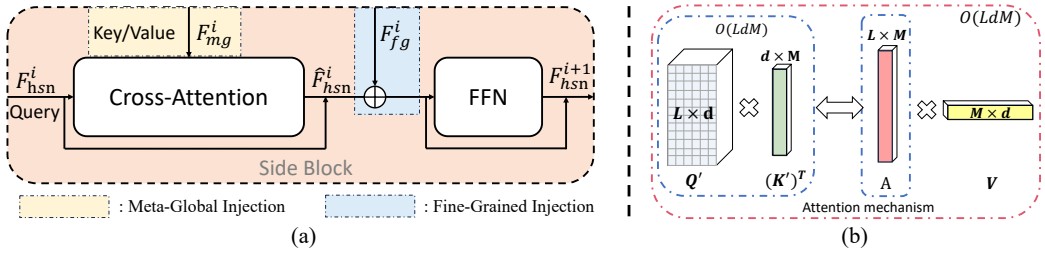

Figure 6: **Side Block.** (a) The schematic illustration of the proposed Side Block. (b) Illustration of linear complexity of cross-attention in Side block.

**Meta-Global Injection.** As illustrated in Figure 6(a), we utilize the multi-scale feature from HSN as the $query(Q)$ matrix and employ meta-global tokens as the $key(K)$ and $value(V)$ matrices for performing cross attention. This process is defined as follows:

$$((Q_{hsn})(K_{mg})^T)V_{mg} = AV_{mg} \tag{2}$$

where $Q_{hsn} \in \mathbb{R}^{L \times d}$, $(K_{mg})^T \in \mathbb{R}^{d \times M}$, and $V_{mg} \in \mathbb{R}^{M \times d}$. Here, $L$ denotes the length of the multi-scale input sequence, $M$ represents the length of the meta-global tokens, and $d$ signifies the feature dimension. This approach affords us the advantage of a computation complexity

of $O(LdM)$. It is noteworthy that both $d$ and $M$ are significantly smaller compared to the input sequence length, $L$ ($d, M \ll L$). Consequently, we can judiciously omit $d$ and $M$, culminating in a computation complexity of $O(L)$ (Figure 6(b)). It allows us to effectively inject global priors into the side network, while also reducing the computational complexity of attention to linear, significantly enhancing the training and inference efficiency of the HSN.

**Fine-Grained Injection.** After cross-attention, we obtain the output feature $\hat{F}_{hsn}^i$, which can be written as follows:

$$\hat{\mathcal{F}}_{hsn}^i = \mathcal{F}_{hsn}^i + \text{CrossAttention}(\mathcal{F}_{hsn}^i, \mathcal{F}_{mg}^i), \tag{3}$$

where $i$ denotes $i$-th block in HST and ViT. Next, we incorporate the fine-grained branch $F_{fg}^i$ into the Side block. Specifically, we perform an element-wise addition of the obtained $\hat{F}_{hsn}^i$ and $F_{fg}^i$ after the cross-attention layer. Subsequently, a feed-forward network (FFN) is applied for further feature modeling. This procedure can be represented as follows:

$$F_{hsn}^{i+1} = \hat{F}_{hsn}^i + F_{fg}^i + \text{FFN}(\hat{F}_{hsn}^i + F_{fg}^i) \tag{4}$$

where the generated feature $F_{hsn}^{i+1}$ will be used as the input of the next Side block.

## 4 EXPERIMENTS

### 4.1 EXPERIMENTAL SETTINGS

**Pre-trained backbone.** For a fair comparison, we adopt the plain Vision Transformer (ViT) (Dosovitskiy et al.) and mainly select ViT-B/16 model pre-trained on ImageNet-21K (Deng et al., 2009) as the initialization for fine-tuning for all downstream tasks. Other details including more benchmarks and different pre-trained approaches are provided in the Appendix B.

**Baseline methods.** We first compare our method with the two basic fine-tuning methods: ($i$) full fine-tuning, where all parameters of the models are updated; ($ii$) linear probing, where only the parameters of the task head are updated. We also compare our method with recent SOTA PETL methods. (Details regarding various PETL methods can be found in the Appendix A.1)

**Downstream tasks.** We evaluate the performance of our HST on both image recognition tasks and dense prediction tasks to confirm its effectiveness. Due to ViT producing feature maps at a single scale (e.g., 1/16th), it could not be adapted to work with a feature pyramid network (FPN) (Lin et al., 2017). Therefore, we follow (Li et al., 2021) to either upsample or downsample intermediate ViT feature maps by placing four resolution-modifying modules to adapt the single-scale ViT to the multi-scale FPN. In this way, similar to recognition tasks, we only need to train the newly added parameters and specific-task head, enabling us to achieve parameter-efficient transfer learning for dense prediction tasks. We provide detailed description in the Appendix A.2.

| | Natural | | | | | | | Specialized | | | | Structured | | | | | | | | | |
|---|---|---|---|---|---|---|---|---|---|---|---|---|---|---|---|---|---|---|---|---|---|
| Method | CIFAR-100 | Caltech101 | DTD | Flowers102 | Pets | SVHN | Sun397 | Camelyon | EuroSAT | Resisc45 | Retinopathy | Clevr/count | Clevr/distance | DMLab | KITTI/distance | dSprites/loc | dSprites/ori | SmallNORB/azi | SmallNORB/ele | Average (%) | Params. (M) |
| Full fine-tuning (2022) | 68.9 | 87.7 | 64.3 | 97.2 | 86.9 | 87.4 | 38.8 | 79.7 | 95.7 | 84.2 | 73.9 | 56.3 | 58.6 | 41.7 | 65.5 | 57.5 | 46.7 | 25.7 | 29.1 | 65.57 | 85.84 |
| Linear probing (2022) | 63.4 | 85.0 | 63.2 | 97.0 | 86.3 | 36.6 | 51.0 | 78.5 | 87.5 | 68.6 | 74.0 | 34.3 | 30.6 | 33.2 | 55.4 | 12.5 | 20.0 | 9.6 | 19.2 | 52.94 | 0.04 |
| Adapter (2019) | 74.1 | 86.1 | 63.2 | 97.7 | 87.0 | 34.6 | 50.8 | 76.3 | 88.0 | 73.1 | 70.5 | 45.7 | 37.4 | 31.2 | 53.2 | 30.3 | 25.4 | 13.8 | 22.1 | 55.82 | 0.27 |
| Bias (2021) | 72.8 | 87.0 | 59.2 | 97.5 | 85.3 | 59.9 | 51.4 | 78.7 | 91.6 | 72.9 | 69.8 | 61.5 | 55.6 | 32.4 | 55.9 | 66.6 | 40.0 | 15.7 | 25.1 | 62.05 | 0.14 |
| VPT-Shallow (2022) | 77.7 | 86.9 | 62.6 | 97.5 | 87.3 | 74.5 | 51.2 | 78.2 | 92.0 | 75.6 | 72.9 | 50.5 | 58.6 | 40.5 | 67.1 | 68.7 | 36.1 | 20.2 | 34.1 | 64.85 | **0.11** |
| VPT-Deep (2022) | 78.8 | 90.8 | 65.8 | 98.0 | 88.3 | 78.1 | 49.6 | 81.8 | 96.1 | 83.4 | 68.4 | 68.5 | 60.0 | 46.5 | 72.8 | 73.6 | 47.9 | 32.9 | 37.8 | 69.43 | 0.60 |
| LoRA (2021) | 67.1 | 91.4 | 69.4 | 98.8 | 90.4 | 85.3 | 54.0 | 84.9 | 95.3 | 84.4 | 73.6 | 82.9 | 69.2 | 49.8 | 78.5 | 75.7 | 47.1 | 31.0 | 44.0 | 72.25 | 0.29 |
| NOAH (2022) | 69.6 | 92.7 | 70.2 | 99.1 | 90.4 | 86.1 | 53.7 | 84.4 | 95.4 | 83.9 | 75.8 | 82.8 | 68.9 | 49.9 | 81.7 | 81.8 | 48.3 | 32.8 | 44.2 | 73.20 | 0.36 |
| AdaptFormer-64 (2022a) | 70.6 | 92.9 | 72.2 | **99.6** | 91.3 | 86.9 | **55.4** | **88.5** | **96.6** | 87.1 | **76.9** | 78.5 | 62.1 | 51.9 | 81.2 | 74.6 | 52.5 | 31.5 | 39.4 | 73.10 | 1.26 |
| SSF (2022) | 69.0 | 92.6 | **75.1** | 99.4 | **91.8** | 90.2 | 52.9 | 87.4 | 95.9 | 87.4 | 75.5 | 75.9 | 62.3 | **53.3** | 80.6 | 77.3 | 54.9 | 29.5 | 37.9 | 73.10 | 0.24 |
| TOAST (2023) | **82.1** | 90.5 | 70.5 | 98.7 | 89.7 | 71.9 | 53.3 | 84.3 | 95.5 | 85.5 | 74.2 | 75.4 | 60.8 | 44.7 | 77.5 | 73.9 | 47.5 | 24.5 | 33.7 | 70.20 | 14.0 |
| EXPRES (2023) | 78.0 | 89.6 | 68.8 | 98.7 | 88.9 | 81.9 | 51.9 | 84.8 | 96.2 | 80.9 | 74.2 | 66.5 | 60.4 | 46.5 | 77.6 | 78.0 | 49.5 | 26.1 | 35.3 | 72.90 | - |
| HST (ours) | 76.7 | **94.1** | 74.8 | **99.6** | 91.1 | **91.2** | 52.3 | 87.1 | 96.3 | **88.6** | 76.5 | **85.4** | 63.7 | 52.9 | 81.7 | 87.2 | 56.8 | 35.8 | 52.1 | **76.00** | 0.78 |

Table 1: Performance comparisons on the VTAB-1k benchmark with ViT-B/16 models.

## 4.2 Classification On VTAB-1K Benchmark

As shown in Table 1, we compare HST with the state-of-the-art PETL methods on ViT across all three splits of VTAB-1k, where the first column is the average accuracy on the 19 downstream tasks and the second column shows the average of tunable parameters. Specifically, HST achieves an average accuracy of 76.00%, outperforming the full fine-tuning on 19 out of 19 tasks with only additional 0.9% of the backbone parameters. Furthermore, HST surpasses SSF, LoRA, AdaptFormer, and NOAH by +2.9%, +3.75%, +2.9% and +2.8% respectively. Compared to existing approaches, HST demonstrates superior performance on VTAB-1K, especially with notable improvements of 6.9%, 5.4%, and 7.9% on Clever/Count, dSprites/loc, and SmallNORB/ele respectively. These results strongly validate the effectiveness and parameter efficiency of our proposed HST method.

## 4.3 Object Detection And Instance Segmentation

As shown in Table 2, regardless of the detector used, existing PETL methods still exhibit a significant performance gap compared to the full-tuning. This disparity stems from the fundamental differences between classification tasks and dense prediction tasks, highlighting the ineffectiveness of PETL techniques in transfer learning for the latter. However, our HST breaks through this performance limit. When training Mask R-CNN with $3\times$ schedule, our HST demonstrates only 1.0 $AP^b$ decrease and achieves equal performance in $AP^m$ compared to full-tuning. Additionally, HST yields a 1.0 $AP^b$ and 1.0 $AP^m$ improvement over full fine-tuning in Cascade Mask R-CNN with $3\times$ schedule, while only exhibiting a 0.6 $AP^b$ decrease compared to full-tuning method in ATSS. These encouraging results indicate that our method enhances transfer robustness and even enables ViT models to achieve superior performance.

| Method | #Param (M) | Mask R-CNN $1\times$ schedule | | | | | | Mask R-CNN $3\times$+MS schedule | | | | | |
|---|---|---|---|---|---|---|---|---|---|---|---|---|---|
| | | $AP^b$ | $AP^b_{50}$ | $AP^b_{75}$ | $AP^m$ | $AP^m_{50}$ | $AP^m_{75}$ | $AP^b$ | $AP^b_{50}$ | $AP^b_{75}$ | $AP^m$ | $AP^m_{50}$ | $AP^m_{75}$ |
| Full fine-tuning (2022) | 113.6 | 43.1 | 65.9 | 46.8 | 39.5 | 62.9 | 42.1 | 45.1 | 67.2 | 48.9 | 40.5 | 63.9 | 43.0 |
| Linear probing (2022) | 27.8 | 22.1 | 43.5 | 20.0 | 22.6 | 41.1 | 22.1 | 25.0 | 47.3 | 23.9 | 24.9 | 44.9 | 24.6 |
| VPT-deep (2022) | 28.4 | 31.1 | 55.0 | 31.1 | 30.5 | 52.0 | 31.1 | 33.4 | 57.4 | 34.3 | 32.2 | 54.0 | 33.3 |
| AdaptFormer (2022a) | 29.0 | 32.8 | 57.4 | 33.4 | 32.2 | 54.3 | 33.1 | 36.7 | 61.6 | 38.5 | 35.1 | 58.1 | 36.6 |
| SSF (2022) | 28.0 | 35.6 | 60.2 | 37.4 | 34.4 | 57.0 | 36.0 | 36.5 | 60.6 | 38.4 | 34.8 | 57.6 | 36.3 |
| LoRA (2021) | 28.4 | 36.2 | 60.9 | 37.5 | 35.0 | 57.9 | 36.5 | 39.3 | 64.1 | 41.6 | 37.1 | 60.6 | 39.1 |
| HST (**ours**) | 30.6 | **40.5** | **64.4** | **43.2** | **38.0** | **61.1** | **40.0** | **44.1** | **67.0** | **47.8** | **40.5** | **64.0** | **43.2** |

| Method | Cascade Mask R-CNN $3\times$ +MS | | | | | | | ATSS $3\times$+MS | | | |
|---|---|---|---|---|---|---|---|---|---|---|---|
| | #Param | $AP^b$ | $AP^b_{50}$ | $AP^b_{75}$ | $AP^m$ | $AP^m_{50}$ | $AP^m_{75}$ | #Param | $AP^b$ | $AP^b_{50}$ | $AP^b_{75}$ |
| Full fine-tuning (2022) | 151.4M | 48.7 | 68.1 | 52.3 | 42.2 | 65.1 | 45.4 | 101.3M | 46.7 | 67.2 | 50.1 |
| Linear probing (2022) | 65.6M | 35.9 | 55.3 | 38.5 | 31.4 | 52.2 | 32.2 | 15.6M | 26.0 | 43.9 | 26.4 |
| VPT-deep (2022) | 66.2M | 42.2 | 62.1 | 45.4 | 37.1 | 59.2 | 39.1 | 16.1M | 35.4 | 55.0 | 37.6 |
| AdaptFormer (2022a) | 66.8M | 45.1 | 65.3 | 48.6 | 39.2 | 62.4 | 41.5 | 16.7M | 38.4 | 58.9 | 40.9 |
| SSF (2022) | 65.6M | 44.2 | 64.2 | 47.8 | 38.6 | 61.0 | 41.0 | 15.8M | 37.8 | 57.8 | 40.4 |
| LoRA (2021) | 66.2M | 46.9 | 67.3 | 50.6 | 40.8 | 64.3 | 43.4 | 16.2M | 41.1 | 62.1 | 44.1 |
| HST (**ours**) | 68.4M | **49.7** | **69.0** | **54.1** | **43.2** | **66.2** | **46.9** | 18.5M | **46.0** | **65.8** | **49.8** |

Table 2: **Object detection and instance segmentation with three main detectors which are Mask R-CNN, Cascade Mask R-CNN and ATSS on COCO val2017.** $AP^b$ and $AP^m$ represent box AP and mask AP, respectively. "MS" means multi-scale training.

## 4.4 Semantic Segmentation

In Table 3, we provide semantic segmentation results in terms of mIoU, both with and using multi-scale (MS) techniques for comparison. Our HST demonstrates impressive performance, achieving a mIoU of 46.5 and 47.3 with MS when integrated with UperNet, surpassing other PETL methods by a minimum margin of 0.5 mIoU, while maintaining the lowest number of trainable parameters. Furthermore, in Semantic FPN, HST attains state-of-the-art results, boasting mIoU scores of 44.3 and 45.0 with MS. However, although HST achieves the best performance, it is evident that we still have room for improvement for segmentation tasks when compared to full fine-tuning. This underscores the ongoing challenge we face and the progress yet to be made.

| Method | Crop Size | Semantic FPN 80k | | | UperNet 160k | | |
|---|---|---|---|---|---|---|---|
| | | #Param | mIoU | +MS | #Param | mIoU | +MS |
| Full fine-tuning (Jia et al., 2022) | 512×512 | 97.7M | 46.0 | 47.2 | 127.0M | 49.5 | 50.8 |
| Linear probing (Jia et al., 2022) | 512×512 | 11.9M | 34.2 | 36.5 | 41.2M | 37.1 | 39.1 |
| VPT-deep (Jia et al., 2022) | 512×512 | 12.5M | 41.5 | 41.4 | 41.8M | 44.0 | 46.1 |
| AdaptFormer (Chen et al., 2022a) | 512×512 | 13.1M | 42.8 | 43.0 | 42.4M | 43.4 | 44.6 |
| SSF (Lian et al., 2022) | 512×512 | 12.1M | 44.2 | 44.6 | 41.4M | 44.9 | 46.8 |
| LoRA (Hu et al., 2021) | 512×512 | 12.5M | 44.0 | 44.9 | 41.8M | 44.9 | 46.4 |
| HST (ours) | 512×512 | 14.7M | **44.3** | **45.0** | 39.9M | **46.5** | **47.3** |

Table 3: **Semantic segmentation on the ADE20K val.** Semantic FPN (Kirillov et al., 2019) and UperNet (Xiao et al., 2018) are used as segmentation frameworks. "MS" means multi-scale testing.

## 4.5 EFFICIENCY ANALYSIS

To validate the inference and training efficiency of our method, we show the computational cost of HST in Figure 7 and 8. We conduct a comparative analysis of the training costs associated with various PETL methods. All results are the average of 100 trials, assessed using V100 GPUs. Our observations reveal that in the image classification benchmark, HST exhibits a training memory requirement similar to that of VPT (with 64 prompts), yet less than SSF and full fine-tuning methods, maintaining the highest accuracy at 76%. In the case of dense prediction benchmarks, HST exhibits a training time requirement comparable to that of SSF, albeit slightly more than AdaptFormer and LoRA. In terms of training memory, all these PETL methods exhibit closely aligned profiles, which are lower than those of full

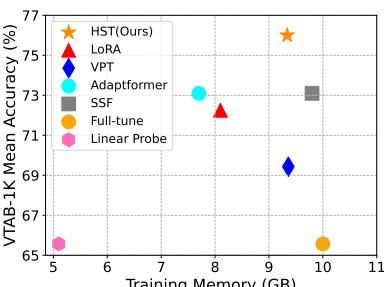

Figure 7: Comparison between different PETL methods over VTAB-1K benchmark.

fine-tuning. Furthermore, it is worth noting that our HST demonstrates comparable inference speeds with other PETL methods on Mask R-CNN and UperNet. This can be attributed to the lightweight nature of HSN, where the number of feature channels in different stages is much smaller than that in ViT. This reduction in feature channels helps alleviate the computational load on the dense prediction head. (More efficiency analysis can be found in the Appendix C)

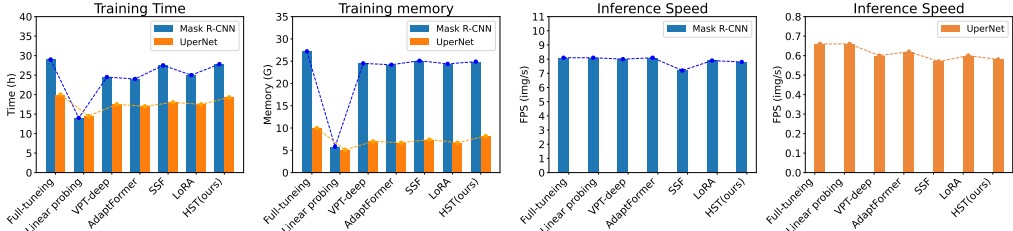

Figure 8: Efficiency comparison over detection and segmentation tasks.

## 4.6 VISUALIZATIONS

As shown in Figure 9, we utilize t-SNE (Van der Maaten & Hinton, 2008) to visualize the feature distributions of HST and other PETL methods, with the goal of evaluating the quality of the generated features. Clearly, our HST exhibits significantly improved feature clustering results. Additionally, we employ Grad-CAM (Selvaraju et al., 2017) to visualize attention maps. The results illustrate that HST can distinctly emphasize target objects, thus affirming the efficacy of our approach. (More visualizations are shown in the Appendix D)

## 4.7 ABLATION STUDIES

We conduct an ablation study on the HST to investigate the critical factors contributing to its effectiveness, uncovering several intriguing insights. All the ablation studies are carried out using the VTAB-1K validation set and MS COCO with Mask R-CNN 1× schedule.

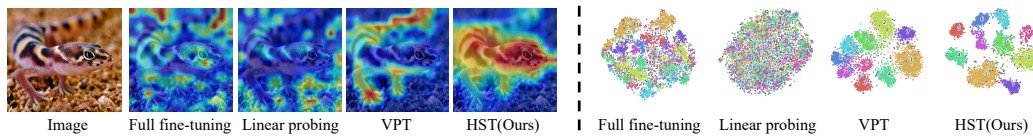

Figure 9: **Left:** Visualization of attention maps. **Right:** t-SNE visualization of various PETL methods applied to the SVHN task within the natural category.

**Number of MetaT** Table 4 showcases the impact of adjusting the number of MetaT tokens on tuning performance. The experimental results indicate that increasing the number of MetaT tokens does not result in a proportionate improvement. Particularly in classification tasks, employing a single MetaT token yields higher accuracy compared to using 4 or 8 tokens. Moreover, an excessive increase in the number of MetaT tokens can potentially have a detrimental impact on the network's efficiency. Therefore, our recommendation is to employ a single MetaT for classification tasks and no more than 8 for dense prediction tasks.

**Ablation for Components.** To investigate the contribution of each key design, we progressively extend the ViT-B with a hierarchical side network to develop our final version of HST. As shown in the first row of Table 5, by directly training the HSN without other approaches, it achieves a classification accuracy of 72.1% in VTAB-1K and 30.0 $AP^b$ and 29.2 $AP^m$ in MSCOCO, which regarded as the baseline. Upon applying the LN tuning method, our HST.a exhibits improvements of 2.2%, 2.8 $AP^b$ and 2.3 $AP^m$ over the baseline. Moving on to HST.b, we surprisingly find that linear weight sharing even outperforms the use of multiple linear layers, especially in VTAB-1K. This finding suggests that an excessive number of linear layers is unnecessary within our tuning framework. In addition, we introduce an average pooling operation to generate a global token, which is concatenated with MetaT and used as an injection in the Side block. This modification (HST.c) results in gains of 0.2% in classification accuracy, 1.7 for $AP^b$, and 1.5 for $AP^m$, effectively enhancing the incorporation of global priors into the HSN. Finally, we implement the Fine-Grained Injection in the Side block, leading to remarkable improvements of 0.8%, 5.5 $AP^b$ and 5.0 $AP^m$ improvement, which demonstrates the critical role of intermediate features from the pre-trained backbone. In summary, each of our proposed components proves to be necessary, and their combined contributions result in substantial improvements of 3.9% in classification accuracy, 10.5 for $AP^b$, and 8.8 for $AP^m$.

| $N$ | Mean(%) | $AP^b$ | $AP^m$ |
|---|---|---|---|
| 1 | 76.0 | 39.8 | 37.3 |
| 4 | 75.8 | 40.3 | 37.8 |
| 8 | 75.6 | 40.5 | 38.0 |
| 16 | 76.1 | 40.6 | 38.1 |
| 32 | 76.2 | 40.7 | 38.3 |

Table 4: Number of MetaT.

| Method | Components | | | | #Param | Mean(%) | $AP^b$ | $AP^m$ |
|---|---|---|---|---|---|---|---|---|
| | LN-Tuning | Weight-Sharing | GlobalT | FG Injection | | | | |
| ViT-B w/. HSN | | | | | 1.07M | 72.1 | 30.0 | 29.2 |
| HST.a | ✓ | | | | 1.10M | 74.3 | 32.8 | 31.5 |
| HST.b | ✓ | ✓ | | | 0.78M | 75.0 | 32.8 | 31.5 |
| HST.c | ✓ | ✓ | ✓ | | 0.78M | 75.2 | 34.5 | 33.0 |
| **HST (ours)** | ✓ | ✓ | ✓ | ✓ | 0.78M | **76.0** | **40.5** | **38.0** |

Table 5: Ablation studies of key components.

## 5 CONCLUSION

In this paper, we introduce Hierarchical Side-Tuning (HST), a new parameter-efficient transfer learning method designed to effectively adapt large vision Transformer backbones. Our tuning framework incorporates a trainable hierarchical side network, which successfully leverages the intermediate features of the backbone and generates multi-scale features for making predictions. Extensive experiments illustrate that HST consistently outperforms previous state-of-the-art methods on diverse benchmarks, significantly reducing the performance disparity between PETL methods and full fine-tuning in dense prediction tasks. We hope that HST will inspire researchers into developing versatile PETL techniques applicable to a wide range of downstream tasks. Key directions for future work include exploring targeted parallel computation to further improve inference efficiency and designing a unified model for simultaneous multiple visual tasks with different HSNs.

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

# A   DETAILED DESCRIPTIONS FOR THE EVALUATION DATASETS AND METHODS

## A.1   EVALUATION METHODS.

($i$) Full fine-tuning, where all parameters of the models are updated; ($ii$) linear probing, where only the parameters of the task head are updated. We also compare our method with recent SOTA PETL methods. ($iii$) Adapter (Houlsby et al., 2019), where a new adapter structure with up-projection, non-linear function, and down-projection is inserted into the transformer and only the parameters of this new module are updated; ($iv$) Bias (Zaken et al., 2021), where all the bias terms of parameters are updated; ($v$) VPT (Jia et al., 2022), where the prompts are inserted into transformers as the input tokens; ($vi$) LoRA (Hu et al., 2021), adopts an optimized low-rank matrix to the multi-head attention module in the transformer layers; ($vii$) AdaptFormer (Chen et al., 2022a), adopts an optimized new Adapter structure to the FFN module in the transformer layers; ($viii$) SSF (Lian et al., 2022), leverages two learnable vectors to scale and shift the feature map in each transformer operation.

## A.2   DOWNSTREAM DATASETS

• *Image Recognition:* The VTAB-1k benchmark was introduced in a preceding scholarly endeavor by (Zhai et al., 2019), comprising a comprehensive array of 19 tasks across diverse domains. These tasks are stratified into three distinct categories: Natural, encompassing images captured through conventional camera devices; Specialized, involving images procured under specific contexts such as medical and satellite imaging; and Structured, which comprises images synthesized within controlled, simulated environments, primarily exemplified by variations in object proximity. Each task-specific dataset contains 1000 training samples with varying number of samples per class. Model evaluation, in this instance, is predicated on performance metrics computed across the entire test set. We directly resize the image to 224×224, following the default settings in (Zhai et al., 2019).

• *Object Detection and Instance Segmentation:* Our detection experiments are based on MMDetection (Chen et al., 2019) and the MS COCO dataset (Lin et al., 2014). We use 3 mainstream detectors to evaluate our HST, including Mask R-CNN (He et al., 2017), Cascade Mask R-CNN (Cai & Vasconcelos, 2019) and ATSS (Zhang et al., 2020b). Following common practices (Wang et al., 2021), we employ $1\times$ and $3\times$ training schedules with a batch size of 16. We utilize the AdamW (Loshchilov & Hutter, 2017) optimizer with an initial learning rate of $1 \times 10^{-4}$ and a weight decay of 0.05.

• *Semantic Segmentation:* Our semantic segmentation experiments are based on MMSegmentation (Contributors, 2020) and the ADE20K (Zhou et al., 2017) dataset which has 20k and 2k images from 150 categories for training and validation. We take Semantic FPN (Kirillov et al., 2019) and UperNet (Xiao et al., 2018) as the basic frameworks. For Semantic FPN, we adopt the same settings as PVT (Wang et al., 2021) and train the models for 80k iterations. As for UperNet, we adhere to the Swin Transformer's (Liu et al., 2021) settings and train it for 160k iterations. We employ the same approach as used in detection to endow ViT with the capability to generate multi-scale feature outputs.

# B   MORE RESULTS OF ADDITIONAL BENCHMARKS AND DIFFERENT PRE-TRAINED APPROACHES

Following VPT (Jia et al., 2022), we utilize four Fine-Grained Visual Classification (FGVC) datasets to assess the performance of our proposed HST approach. These datasets include CUB-200-2011 (Wah et al., 2011), Oxford Flowers (Nilsback & Zisserman, 2008), Stanford Dogs (Khosla et al., 2011), and Stanford Cars (Gebru et al., 2017). Additionally, we employ the CIFAR-100 (Krizhevsky et al., 2009) dataset as a general image classification benchmark to further confirm the effectiveness of HST. Moreover, to evaluate the adaptability of these parameter-efficient transfer learning (PETL) methods across various pre-training techniques, we predominantly choose the

| Method \ Dataset | Cifar-100 | CUB-200 -2011 | Oxford Flowers | Stanford Dogs | Stanford Cars | Params.(M) |
|---|---|---|---|---|---|---|
| Full fine-tuning | 93.8 / 88.9 | 87.3 / 83.0 | 98.8 / 90.9 | 89.4 / 84.6 | 84.5 / 91.5 | 85.98 |
| Linear probing | 88.7 / 36.9 | 85.3 / 31.7 | 97.9 / 46.0 | 86.2 / 53.2 | 51.3 / 32.8 | 0.18 |
| Adapter (2019) | 93.3 / 74.9 | 87.1 / 74.0 | 98.5 / 85.0 | 89.8 / 78.4 | 68.6 / 72.5 | 0.41 |
| Bias (2021) | 93.4 / 76.3 | 88.4 / 74.3 | 98.8 / 84.4 | **91.2** / 80.8 | 79.4 / 73.8 | 0.28 |
| VPT-Shallow (2022) | 90.4 / 73.1 | 86.7 / 71.1 | 98.4 / 86.5 | 90.7 / 68.8 | 68.7 / 79.0 | 0.25 |
| VPT-Deep (2022) | 93.2 / 74.2 | 88.5 / 73.3 | 99.0 / 87.4 | 90.2 / 71.5 | 83.6 / 81.9 | 0.85 |
| HST (**ours**) | **93.6 / 79.7** | **89.2 / 78.7** | **99.6 / 91.2** | 89.5 / **86.4** | **88.2 / 83.7** | 0.78 |

Table 6: Performance comparisons on Cifar-100 and four FGVC datasets with ViT-B/16 models pre-trained on **ImageNet-21K / MAE**.

ViT-B/16 (Dosovitskiy et al.) model, pre-trained on ImageNet-21K[1], and MAE[2] (He et al., 2022a) as the initialization for fine-tuning.

As the results shown in Table 6, under ImageNet-21K pre-training, HST achieves comparable performance on the CIFAR-100 dataset (93.6% vs. 93.8%) and surpasses full fine-tuning on four FGVC datasets. Furthermore, HST utilizes fewer trainable parameters compared to VPT-Deep in most of these datasets (0.78M vs. 0.85M). In the case of MAE pre-training, it is evident that other PETL methods exhibit subpar performance, with most of them significantly falling below the level of full fine-tuning. This indicates their limited adaptability across different pre-training methods. In contrast, HST not only outperforms full fine-tuning on certain datasets but also maintains a minimal performance gap on others. This underscores the versatility and effectiveness of HST across a wide range of pre-training approaches

## C    ADDITIONAL EFFICIENCY ANALYSIS

To evaluate the inference efficiency of various PETL methods, we present GPU latency in this section. In Table 7, we offer a comparison of inference speeds on the classification benchmark. Notably, all PETL methods introduce varying degrees of inference slowdown. We have observed that in the case of single-batch inference, factors such as network depth and the inclusion of additional network units have a more pronounced impact on GPU latency. Conversely, for multi-batch inference, the critical factor affecting GPU latency is the number of tokens input to the Transformer. For instance, when employing a batch size of 1 in VPT, the latency remains nearly equivalent to that of full fine-tuning. However, with batch sizes of 32 or 128, there is a significant impact on latency. Regarding HST, due to the incorporation of a hierarchical side network, it demands greater computational resources, which consequently results in slightly slower inference speeds compared to other PETL methods. However, these results stem from a serial calculation process rather than a parallel one. Specifically, when the output of a ViT block is computed, it awaits the completion of the corresponding Side block calculation before proceeding to the next one. In reality, there is no need for the calculation of the subsequent ViT block to wait for the Side block's computation, and ViT and HSN calculations can be executed in parallel. Hence, by employing targeted parallel computing methods, the inference speed of HST can be substantially enhanced

## D    MORE VISUALIZATIONS

### D.1    FEATURE QUALITY

We employ t-SNE to visualize the feature distributions of HST and other baseline methods, aiming to assess the quality of the generated features. These features are extracted from three distinct tasks: Caltech101, EuroSAT, and KITTI-Dist, each representing a different category. We utilize a ViT-B/16

---

[1] https://github.com/rwightman/pytorch-image-models/releases/download/v0.1-vitjx/jx_vit_base_patch16_224_in21k-e5005f0a.pth.

[2] https://dl.fbaipublicfiles.com/mae/pretrain_vit_base.pth.

| Method | FLOPs (G) | GPU latency (imgs/sec) | | |
|---|---|---|---|---|
| | | bs=1 | bs=32 | bs=128 |
| Full fine-tuning (2022) | 16.9 | 118.0 | 302.8 | 306.0 |
| VPT-deep (2022) | 22.3 | 116.0 | 216.5 | 229.6 |
| AdaptFormer (2022a) | 17.1 | 101.0 | 291.5 | 296.2 |
| SSF (2022) | 16.9 | 93.4 | 269.0 | 274.5 |
| LoRA (2021) | 17.0 | 88.6 | 290.3 | 294.2 |
| HST (ours) | 17.5 | 70.5 | 240.2 | 248.1 |

Table 7: **Efficiency comparison.** We use ViT-B/16 as the backbone.The inference speed is defined by images per second (imgs/sec). All results are the average of 100 trials.

model pretrained on the ImageNet-21K dataset as the basis for feature extraction. In Figure 10, it is evident that both linear fine-tuning and full fine-tuning methods tend to produce mixed features. In comparison, our HST demonstrates superior feature clustering results when contrasted with VPT and LoRA. This observation further validates that our HST enhances the ability of ViT-B to generate more distinguishable representations while requiring fewer learnable parameters.

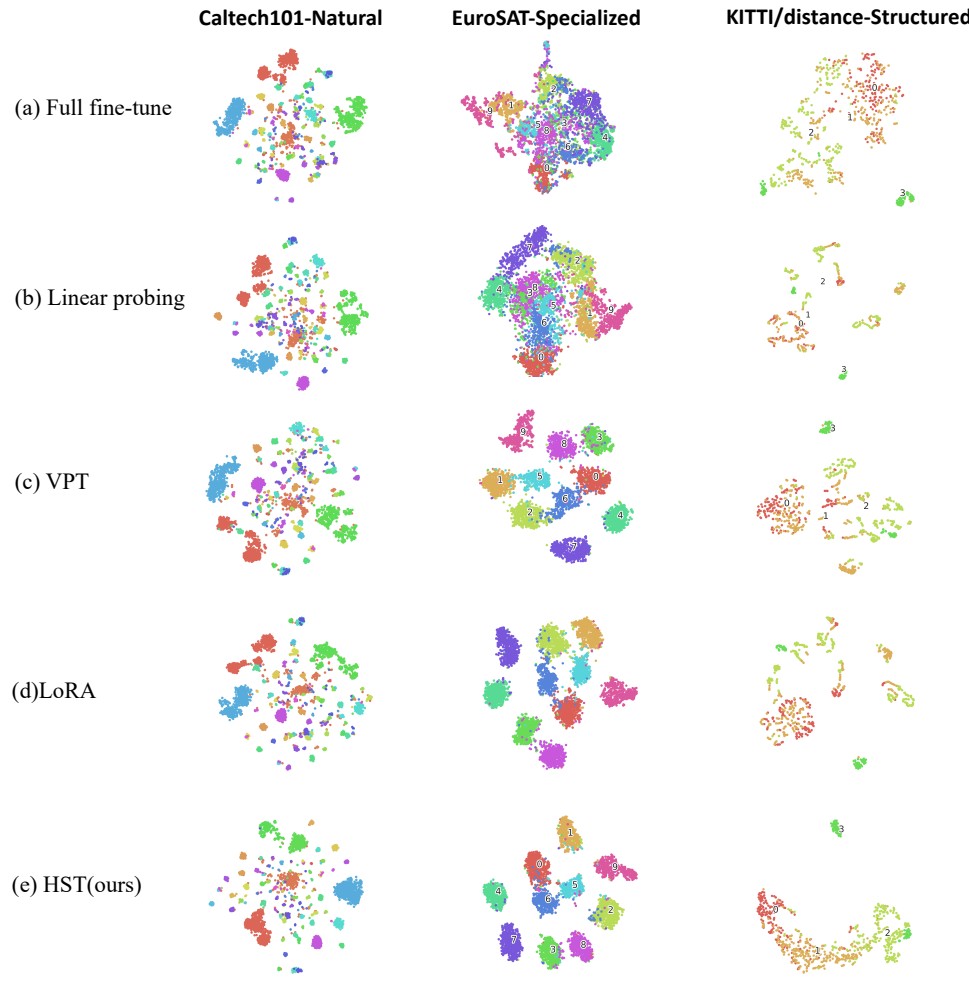

Figure 10: **t-SNE (Van der Maaten & Hinton, 2008) visualization of different PETL methods** on 3 tasks from each VTAB-1k's category. We extract the final classification features for t-SNE visualizations

### D.2 ATTENTION MAP

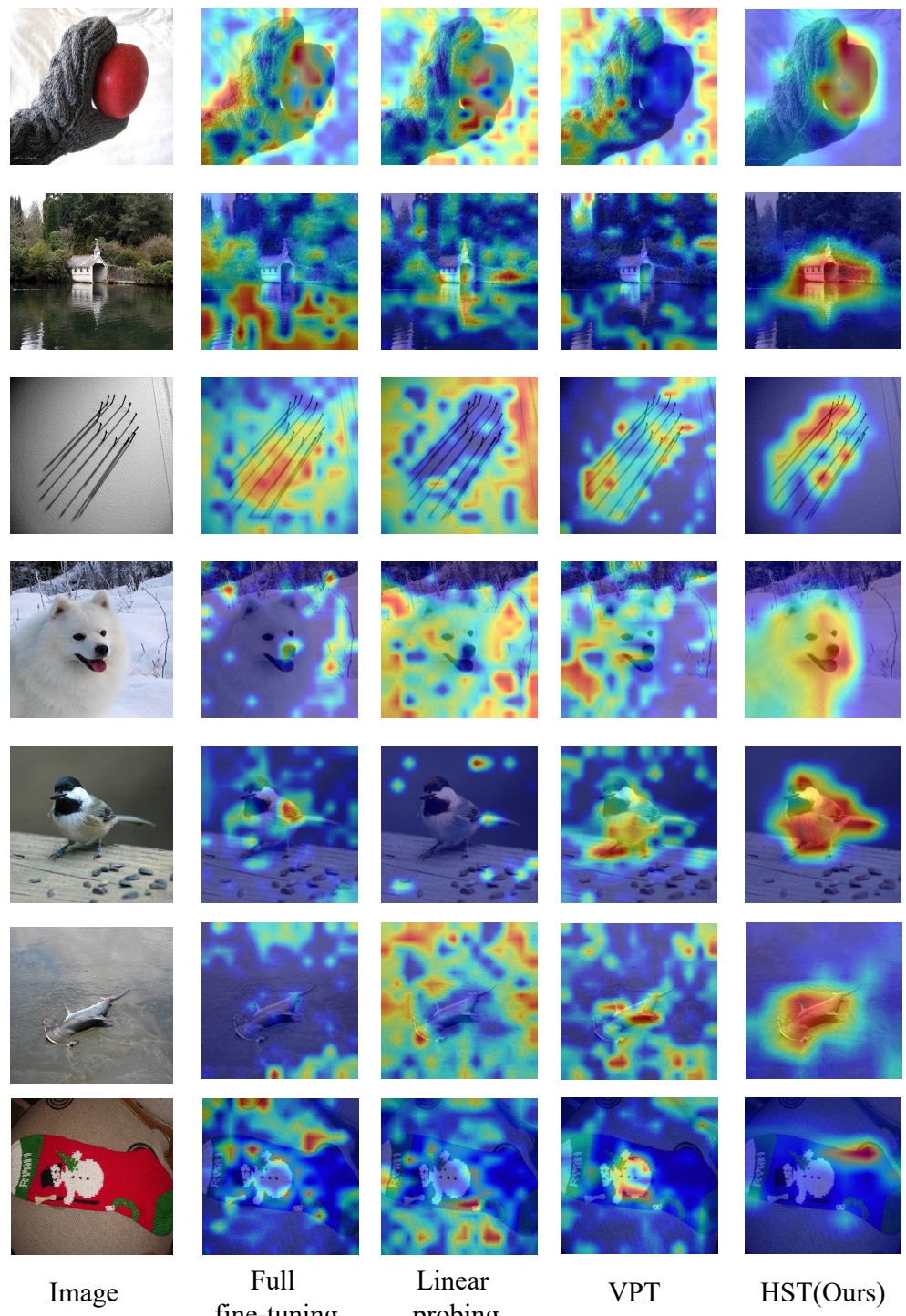

Figure 11: **Visualization results.** We utilize Grad-CAM (Selvaraju et al., 2017) to visualize attention maps on the ImageNet-1k validation set. Each column presents the RGB image, full fine-tuning, linear probing, VPT-Deep, and our HST. The results illustrate that HST can distinctly emphasize target objects, thus affirming the efficacy of our approach.

