# OpenReview forum: "Hierarchical Side-Tuning for Vision Transformers"
_ICLR.cc/2024/Conference — Submitted to ICLR 2024_

### Official Review · Reviewer_cBa6 · 2023-10-29

**Soundness:** 3 good
**Presentation:** 4 excellent
**Contribution:** 3 good
**Rating:** 6
**Confidence:** 4

**Summary:**

The paper proposes a Parameter Efficient Transfer Learning (PETR) approach Hierachical Side-Tuning (HST) where a backbone network is frozen, and activations are injected into a smaller "side" network. The architecture is targeted mainly at dense prediction tasks (like segmentation), where there is a large gap between PETR and full fine-tuning.

The approach is relatively simple (a strength), and the results are compelling. The authors evaluate the approach on both dense prediction (COCO) and the classification-style tasks (VTAB), in a reasonably extensive set of evaluations. The experiments are a strong point of the paper.

Overall I gave this paper a 6, but with some additional experimental validation I would consider increasing my rating.

**Strengths:**

### Positioning
Generally I found this to be a well-presented paper. Easy to follow and well-written.

I agree with the authors that most of the transfer learning literature is aimed at classification or text-based transfer, and PETR for dense prediction is an understudied problem. So I like the setup of the paper.

One advantage of HST is that, similar to other side-tuning approaches like LST and ST, “gradient computation for the trainable parameters does not requrie backpropagation through the large pre-trained backbone model.” The authors don't mention this, but Fig 2 kind of makes this point.


### Experiments:

- Good that the authors included an ablation study on the model components
- Good experiments and comparisons in VTAB in Table 1
- Good experiment on multiple architectures using standard learning rate schedules in table 2

**Weaknesses:**

### Related work
The literature review is generally good, but it would be helpful to discuss the relationship to other side-tuning approaches for dense prediction -- e.g. [Side Adapter Network for Open-Vocabulary Semantic Segmentation
](https://arxiv.org/abs/2302.12242).

### Experiments:

Generally the experiments were well-done, but I think the paper would be stronger with some additional experiments:

- I’d love to see a scaling analysis. E.g. what happens as you scale up the size of the HST network? I.e. compare the performance as you scale up the training data, network size — and compare the performance to analogous settings of LoRA
- The ablation study on VTAB is well-done, but since the paper positions HST as aimed mainly at dense prediction it would be helpful to see a similar analysis on COCO — or at least the current ablation study broken down over classification/structured/specialized (as in Table 1)
- The grad-cam visualizations were nice, but were these cherry picked? The results were unsubstantiated with numbers (e.g. object localization via grad-cam) and weren’t really discussed in the paper. I’d either add a discussion or remove the visualizations.

**Questions:**

Questions:
- I am curious how well-tuned were the baselines for LoRA, etc. Were these coped from existing papers, or did these have a similar HP search, compared to HST?
- For COCO, why use a ResNext and not a ViT/SWIN-based method that performs better? SWIN is hierarchical, or you could use the same linear interp strategy for ViT
- Why do you think the method outperforms full FT for VTAB, but not as strong for ObjDet/SemSeg? Is this simply a result of COCO being a large dataset? The scaling experiment above might help answer this.
- Figure 2: Why is the conv stem necessary? What happens when it is excluded? How is the architecture adapted for classification tasks in VTAB?

---

> ### Author Response · Authors · 2023-11-19
> **Response to Reviewer cBa6 (1/2)**
>
> Thank you for your constructive comments and suggestions. Below, we have provided a point-by-point response to address your comments and concerns.
>
> >**w1: Differences among HST, SAN and ViT-Adapter.**
>
> Please refer to the **(3) Differences among HST, ST, LST, SAN, and ViT-Adapter** section in the general response.
>
> >**w2: Scaling analysis.**
>
> | Mask R-CNN 1x schedule (ViT-L) | Params(M) | APb  | APb(50) | APb(75) | APm  | APm(50) | APm(75) |
> |-|:-:|:-:|:-:|:-:|:-:|:-:|:-:|
> | Full-tuning| 337.3| 45.7 | 68.9| **49.4** | 41.5 | 65.6| **44.6**|
> | Head-tuning| 33.6| 31.6 | 56.4 | 32.0| 31.3 | 53.3 | 32.5|
> | LoRA-64| 39.84| 44.8 | 68.8 | 48.9| 41.2 | 65.3 | 44.0|
> | HST| 39.62| **45.7** | **69.1**| 49.2 | **41.6** | **65.6** | 44.4|
>
> We can observe that HST performs more satisfactorily when using larger models like ViT-L. There is a performance gap of 2.6 AP_b between HST and full finetune on the base model, while achieving the same or even better performance on the large model. Additionally, the LoRA method also demonstrates further improvement in performance with larger models, although it still remains slightly lower than full finetune by approximately 1.0 AP_b.
>
> >**w3: It would be helpful to see a similar analysis on COCO — or at least the current ablation study broken down over classification/structured/specialized.**
>
> In the ablation study, we provided both the average precision on VTAB and the values of AP_b and AP_m using Mask R-CNN on COCO. Here, we present more detailed experimental results on VTAB.
>
> |Ablation MetaT (N) | Natural | Specialized | Structured |
> |:-:|:-:|:-:|:-:|
> | 1 | 82.83 | 87.13 | 64.45 |
> | 4 | 82.88 | 86.08 | 63.66 |
> | 8 | 82.78 | 87.05 | 64.53 |
> | 16 | 82.97 | 87.75 | 64.58 |
> | 32 | 83.02 | 87.70 | 64.71 |
>
> |Method|LN-Tuning|Weight-Sharing|GlobalT|FG Injection|Natural|Specialized|Structured|
> |:-:|:-:|:-:|:-:|:-:|:-:|:-:|:-:|
> |ViT-B w/. HSN| |  |  | | 81.44 | 85.43 | 57.16 |
> |HST.a| +|  |  | | 82.84 | 86.17 | 61.90 |
> |HST.b| + | + | |  | 82.74 | 86.67 | 62.26 |
> |HST.c| + | + |+ |  | 82.74 | 86.67 | 62.26 |
> |HST| +|+ |+|+| 82.83 | 87.13 | 64.45 |
>
> >**w4: Questions about the grad-cam visualizations.**
>
> We can confidently state that these results have not been cherry-picked. In fact, due to limitations in the length of the manuscript, we were only able to temporarily provide a set of visualizations in the main text. Nevertheless, more visualizations are available in Figure 11 of the appendix. It can be observed that the visualization outcomes of Grad-CAM are as expected, owing to the hierarchical architecture of HSN employing convolutional embeddings and modeling fine-grained features in the shallow stages. Consequently, the model naturally has the ability to focus on detailed foreground representations.
>
> >**Q1: I am curious how well-tuned were the baselines for LoRA, etc. Were these coped from existing papers, or did these have a similar HP search, compared to HST?**
>
> While numerous PETL methods primarily concentrate on image classification tasks, the experimental results for these methods in image classification are widely available in multiple papers. Our paper's VTAB results are consistent with those reported. However, when it comes to dense prediction tasks, it becomes challenging to find relevant and dependable results in existing literature. Therefore, we took the initiative to independently replicate the results of all relevant methods in dense prediction tasks. We ensured that all experiments were conducted with the same set of training hyperparameters to maintain consistency across the evaluations.
>
> >**Q3: Why do you think the method outperforms full FT for VTAB, but not as strong for ObjDet/SemSeg?**
>
> Under the experimental setup of the VTAB benchmark with only 1000 training images, full fine-tuning can easily lead to overfitting. However, when we transition to general image classification datasets such as CIFAR-100 or ImageNet-1k (as shown in Table 6 in our paper's appendix or Table 5 in SSF[1]), we observe that full fine-tuning still maintains optimal performance. Therefore, the size of the training dataset plays a significant role.
>
> Additionally, it is important to note that detection and segmentation tasks differ fundamentally from classification tasks. Dense prediction tasks require models to discern features at various granularities, and simply adding a few trainable parameters to the backbone does not enhance the model's ability to model multi-scale features. As a result, the performance may be inferior.
>
> Finally, as mentioned in the scaling analysis, based on the growth trend, we believe that PETL methods will exhibit more powerful performance as the pre-trained model becomes larger. Hence, the size of the model is also one of the influencing factors.
>
> [1] Lian D, Zhou D, Feng J, et al. Scaling & shifting your features: A new baseline for efficient model tuning[J]. Advances in Neural Information Processing Systems, 2022, 35: 109-123.

---

> ### Author Response · Authors · 2023-11-19
> **Response to Reviewer cBa6 (2/2)**
>
> >**Q4: Why is the conv stem necessary? What happens when it is excluded? How is the architecture adapted for classification tasks in VTAB?**
>
> Firstly, the conv stem serves a similar purpose to the patch embedding in popular pyramid-style vision Transformer models. It enables downsampling to reduce the number of input tokens by fourfold. The conv stem consists of two 3x3 convolutional layers with a stride of 2, which introduces spatial prior knowledge related to the image. In our experiments, we observed that the conv stem performs better in terms of both performance and efficiency compared to directly using a 4x4 patch embedding with a stride of 4.
>
> If the conv stem layer is removed, we would directly model an H x W feature map. In this scenario, whether conducting self-attention or cross-attention, the model would face training instability due to the excessively large number of input tokens.
>
> Lastly, when transitioning to image classification tasks, there is no need to utilize intermediate [1/4, 1/8, 1/16] features. Only the 1/32 output feature requires a global average pooling operation, followed by a linear layer for classification.

---

### Official Review · Reviewer_XCzm · 2023-11-01

**Soundness:** 2 fair
**Presentation:** 3 good
**Contribution:** 3 good
**Rating:** 5
**Confidence:** 4

**Summary:**

This paper is motivated by the observation that current PETL (Parameter-Efficient Transfer Learning) methods cannot compare with fully fine-tuning on dense prediction tasks such as semantic segmentation or object detection.  The paper proposed a new PETL method, called Hierarchical Side-Tuning (HST), which uses a trainable side network that generates multi-scale features and leverages intermediate features from the pre-trained backbone. The paper shows that HST outperforms existing parameter-efficient transfer learning methods on image recognition and dense prediction tasks.

**Strengths:**

The motivate is clear. The proposed method is sensible and the experimental results look very promising.

**Weaknesses:**

1. This paper is motivated by the observation that current PETL methods cannot compare with fully fine-tuning on dense prediction tasks such as semantic segmentation or object detection, and the proposed side tuning method is targeted at solving this problem. Another related work [1] also proposes a vision transformer adapter for dense prediction tasks. What are the key differences between this work and ViT-adapter, in terms of model architecture, training strategy, etc.?

2. The paper emphasizes the importance of a side network which leverages intermediate activations extracted from the backbone and generates multi-scale features to make predictions. However, the proposed algorithm also incorporates several other designs that share similarity with previous work, for example, adding meta token is similar to VPT [2], and LN-tuning is reminiscent of BitFit [3] which tunes the bias in the network. Although the final results are very promising, it is not clear the improvement comes from the side network or other designs. An apple-to-apple comparison to baselines to show that the side network really matters (no LN tuning, not meta token, etc.) is probably important to support the claim that side network is important for dense prediction.

3. I'm confused by the number of tunable parameters. Table 1 indicates #params is only 0.78M. However, if each side block contains a FFN and they are all tunable, the #params should be way more than 0.78M, since each FFN will contain several millions of parameters.  Why is the reported #param so low?  Please elaborate on how #params is obtained and what is the exact architecture design of side blocks.

[1] Chen, Zhe, et al. "Vision transformer adapter for dense predictions." arXiv preprint arXiv:2205.08534 (2022).
[2] Jia, Menglin, et al. "Visual prompt tuning." European Conference on Computer Vision. Cham: Springer Nature Switzerland, 2022.
[3] Zaken, Elad Ben, Shauli Ravfogel, and Yoav Goldberg. "Bitfit: Simple parameter-efficient fine-tuning for transformer-based masked language-models." arXiv preprint arXiv:2106.10199 (2021).

**Questions:**

See Weakness.

---

> ### Author Response · Authors · 2023-11-19
> **Response to Reviewer XCzm**
>
> Thank you for your valuable feedback. Below, we address your comments and provide our responses point by point.
>
> >**w1: Differences between ViT-Adapter and HST.**
>
> Please refer to the **(3) Differences among HST, ST, LST, SAN, and ViT-Adapter** section in the general response.
>
> >**w2: The proposed algorithm incorporates several other designs that share similarity with previous work, VPT and BitFit, and it is not clear the improvement comes from the side network or other designs.**
>
> MetaT and prompt tokens in VPT serve the purpose of introducing trainable tokens into ViT. However, they have distinct capabilities within their respective fine-tuning frameworks. MetaT is designed to facilitate efficient computation, using linear complexity for cross-attention within HSN while integrating acquired knowledge into HSN. In contrast to VPT, our approach eliminates the need to manually determine the optimal number of tokens for each task, which often requires a large quantity. Additionally, LN-tuning is employed to reduce the distribution gap between MetaT and the output features of image patch tokens, enabling efficient meta-global/fine-grained injection.
>
> Furthermore, to better illustrate the effects of the side network, we conducted two specific experiments. HST.v0 is without MetaT and LN-Tuning, with cross-attention relying on globalT to serve as the key and value. The second experiment is to add LN-Tuning to LoRA, verifying the effect of integrating different PETL methods. The results are as follows.
>
> | Method | Mean(%) | APb| APm|
> |-|:-:|:-:|:-:|
> |HST.v0|74.1|33.5|32.9|
> |LoRA|72.3|36.2|35.0|
> |LoRA with LN_Tuning|72.9|37.0|36.0|
>
> We can observe that despite the removal of MetaT and LN-Tuning, this method still achieves state-of-the-art performance in classification tasks and surpasses VPT and AdaptFormer in dense prediction tasks. However, the absence of MetaT and LN-Tuning results in the loss of their crucial ability to inject meta-global/fine-grained information, which is the most critical aspect of our framework. On the other hand, even with the integration of LN-Tuning, LoRA does not exhibit a significant improvement in accuracy. Therefore, although MetaT and LN-Tuning may share similarities with other methods, they play distinct roles in our framework, and each component complements the others to achieve optimal performance.
>
> >**w3: Confused by the number of tunable parameters.**
>
> We apologize for the confusion caused by the lack of detailed explanation regarding the exact architecture design. Please refer to our **(2) Detailed Architecture Specifications** provided in the general response, and feel free to ask if you have any further questions.

---

### Official Review · Reviewer_eJkh · 2023-11-05

**Soundness:** 3 good
**Presentation:** 3 good
**Contribution:** 2 fair
**Rating:** 5
**Confidence:** 4

**Summary:**

In this paper, Hierarchical Side-Tuning (HST) is designed for Parameter-efficient Transfer Learning. The HST uses the features from the backbone and generates features for dense prediction with multi-scales. Experiments show HST achieves the SOTA or competitive performance in downstream tasks, like image classification, object detection, and semantic segmentation.

**Strengths:**

1. The paper is easy to follow.
2. The experimental results show significant performance gains for image classification, object detection, and semantic segmentation.

**Weaknesses:**

1. Side-adapter network (SAN) has been used for Open-Vocabulary Semantic Segmentation [1], what's the difference between SAN and the proposed HST?
[1] Side Adapter Network for Open-Vocabulary Semantic Segmentation, CVPR 2023.
2. How to implement the hierarchical dense prediction in Fig. 2?
3. It's not convincing of the analysis of "constraining the number of trainable prompts to a few number". Besides, the ablation study shows continuous growth when the number N increases.
4. 14.4% (76.0% vs. 65.6%)  --> 10.4%

**Questions:**

If it clearly described the novelty of HST compared with SAN, I would change the rating.

---

> ### Author Response · Authors · 2023-11-19
> **Response to Reviewer eJkh**
>
> Thank you for your careful review and valuable feedbacks. In the following section, we address your comments and provide point-by-point responses.
>
> >**w1: Differences between SAN and HST.**
>
> Please refer to the **(3) Differences among HST, ST, LST, SAN, and ViT-Adapter** section in the general response.
>
> >**w2: How to implement the hierarchical dense prediction in Fig. 2?**
>
> In simple terms, by equipping ViT with a trainable side network to enable its multi-scale output capability, it can be viewed as an overarching pyramid-based architectural model. Subsequently, by incorporating neck modules (FPN, ChannelMapper, etc) and any desired detection or segmentation heads, it can effectively accomplish various dense prediction tasks.
>
> >**w3: It's not convincing of the analysis of "constraining the number of trainable prompts to a few number". Besides, the ablation study shows continuous growth when the number N increases.**
>
> The quantity of MetaT in HST plays a crucial role in determining both the transfer performance and the computational complexity of the Side blocks. Increasing the number of MetaT has an intuitive link with performance enhancement, as observed in VPT as well. However, our findings indicate that using just one MetaT is sufficient to yield satisfactory results in image classification transfer tasks. Although a slight performance boost is observed with 32 MetaT, it significantly increases training and inference costs, which is similar in dense prediction tasks. Therefore, to strike a balance between speed and accuracy, it is favorable to choose varying MetaT quantities (not exceeding 8) based on specific tasks as a hyperparameter choice.
>
> >**w4: 14.4% (76.0% vs. 65.6%) --> 10.4%**
>
> Thanks for pointing out this mistake in our paper.  We will carefully review the paper to ensure that such mistakes are corrected in the final version.

---

### Official Review · Reviewer_ExGi · 2023-11-11

**Soundness:** 2 fair
**Presentation:** 3 good
**Contribution:** 3 good
**Rating:** 5
**Confidence:** 4

**Summary:**

This paper proposed a novel parameters efficient tuning method, called HST. It adopts a FPN-like strategy to build a lightweight side network as learnable components for fine-tuning. Experiments show that the proposed method outperforms compared methods on various downstream tasks.

**Strengths:**

1.	The writing is pretty good and easy to understand.
2.	The improvements, especially on dense prediction tasks, are obvious.

**Weaknesses:**

1.	More implement details, like the d of HSN, are not provided. And as shown in Figure 2, the outputs for dense prediction tasks are from the HST. In my opinion, the channel of outputs is d that is very small. Thus, I'm a bit surprised that inputting such low rank features into the decoder can improve performance.
2.	More fair comparison may be needed. The parameters of compared lora/adapter/SSF are less than 50% that of HST. I suggest to add more comparison under the similar learanable parameters.

**Questions:**

In general, multi-scale feature enhance is a widely used way for dense prediction tasks. Especially in MAE related works, like ConvMAE and iTPN, this type of multi-scale feature architectures is useful for plain ViTs. Therefore, the improvement in performance did not bring me too many surprises. In addition, compared to lora based methods, the proposed method surely leads to additional computational costs and is not as flexible as other methods. My main concerns are mentioned in the weakness and other questions are as follows:
1.	Effect of globalT. In the ablation study, globalT shows improvements without GF injection. I would like to know if globalT is still necessary even if GF injection exists.
2.	In ablation study, weight-sharing does not drop the performance. Does this mean that injecting the features of each block is unnecessary? Or, if we reduce the number of side block in each stage and increase their middle channels to keep the similar parameters. How about this design?
3.	The effect of metaT is not proved.
4.	Can the authors provide more discussion about the comparison on Table 1, since the performance is not always the best.
5.	The performance on much large models, like ViT-L or ViT-H.

---

> ### Author Response · Authors · 2023-11-19
> **Response to Reviewer ExGi (1/2)**
>
> Thank you for your valuable feedback. We have carefully considered your comments and provide our responses below.
>
> >**More implement details**
>
> Please refer to our **(2) Detailed Architecture Specifications** given in the general response.
>
> >**More fair comparison**
>
> By adjusting the dimensions from [32, 48, 64, 72] to [24, 24, 32, 32], we were able to achieve similar numbers of learnable parameters. The experimental results presented in the table below clearly demonstrate that our method consistently delivers state-of-the-art performance on VTAB-1K.
>
> | Method| Mean(%) | Params(M) | CIFAR-100 | Caltech101 | DTD  | Flowers102 | Pets | SVHN | Sun397 |
> |:-:|:-:|:-:|:-:|:-:|:-:|:-:|:-:|:-:|:-:|
> |SSF|73.1|0.24|69.0|92.6|**75.1**| 99.4|**91.8**|90.2|52.9|
> |LoRA|72.3|0.29|67.1|91.4|69.4|98.8|90.4|85.3|**54.0**|
> |HST|**74.6**|0.28|**76.2**|**95.1**|74.2|**99.6**|90.1|**90.8**|47.2|
>
> | Patch Camelyon | EuroSAT | Resisc45 | Retinopathy |
> |:-:|:-:|:-:|:-:|
> |87.4|95.9|**87.4**|75.5|
> |84.9|95.3|84.4|73.6|
> |**87.8**|**96.0**| 87.0|**75.9**|
>
> | Clevr/count | Clevr/distance | DMLab | KITTI/distance | dS/loc | dS/ori | SN/azi | SN/ele |
> |:-:|:-:|:-:|:-:|:-:|:-:|:-:|:-:|
> |75.9|62.3| 53.3| 80.6| 77.3 | 54.9|29.5|37.9|
> |82.9|**69.2**|49.8| 78.5| 75.7| 47.1|**31.0**|44.0|
> |**83.8**|61.8|**53.9**|**83.2**|**86.3**|**55.4**|26.2|**46.2**|
>
> >**w1: leads to additional computational costs**
>
> Apart from LoRA-based techniques capable of weight fusion during inference, most mainstream methods add extra computational load (our paper includes relevant analysis). However, our approach offers a potential advantage by accelerating inference speed through optimized parallel computation. Specifically, our method facilitates concurrent computation, allowing independent progress of calculations in both the ViT network and HSN. This means that the HSN can compute simultaneously using various ViT output features obtained during the forward process of ViT.
>
> >**Q1: Effect of globalT**
>
> As depicted in the table below, we performed an additional ablation experiment without GlobalT, focusing solely on FG Injection. We observed a significant improvement in performance solely due to the implementation of FG Injection. However, when combining both FG Injection and GlobalT, we achieved even better results. Importantly, the introduction of GlobalT resulted in negligible overhead.
>
> |Method|LN-Tuning|Weight-Sharing|GlobalT|FG Injection|Mean(%)|AP_b| AP_m|
> |:-:|:-:|:-:|:-:|:-:|:-:|:-:|:-:|
> |HST.c| +| + | + | | 75.2|34.5|33.0|
> |HST.d| + | + | | + | 75.7|39.5|37.1|
> |HST| +|+ |+|+| 76.0|40.5|38.0|
>
> >**Q2: Issues about weight-sharing and number of side block in each stage**
>
> Thank you for your informative comment. We would like to say that the designs you mentioned were indeed part of our HSN early version. In that version, we experimented with setting only one Side block in each stage, resulting in a total of four Side blocks in the HSN. Furthermore, we fused the output features of different ViT blocks within the same stage, along with their respective meta tokens, and input them into the Side Block. However, this design approach did not yield satisfactory results. We speculate that the inadequate performance may be attributed to the insufficient number of Side blocks in effectively modeling the fusion of multi-scale image features and ViT feature information within the HSN. Despite having a larger number of channels, the network was too shallow. This discovery ultimately inspired us to the design of alignment of ViT blocks and Side blocks.
>
> >**Q3:  The effect of metaT is not proved**
>
> |Method| Mean(%)|AP_b|AP_m|
> |-|:-:|:-:|:-:|
> |only GlobalT| 75.3|38.0|36.1|
> |only MetaT| 75.7| 39.5|37.1|
> |MetaT + GlobalT|76.0|40.5|38.0|
>
> As indicated in the table, the performance achieved solely using GlobalT does not surpass that obtained by solely using MetaT. This discrepancy is primarily due to MetaT's ability to extract more enriched feature from each ViT block compared to GlobalT. Additionally, similar to the prompts in VPT, MetaT demonstrates the capability to transfer information to downstream tasks. We apologize the oversight in not providing an analysis of MetaT's effectiveness in the paper. We will provide more details to elaborate on the effectiveness of MetaT.

---

> ### Author Response · Authors · 2023-11-19
> **Response to Reviewer ExGi (2/2)**
>
> >**Q4: Providing more discussion about the comparison on Table 1, since the performance is not always the best.**
>
> Firstly, let's delve into the VTAB-1k benchmark in detail. The VTAB-1k benchmark consists of 19 tasks spanning various domains, including Natural (images from standard cameras), Specialized (context-specific domains like medical and satellite imaging), and Structured (synthesized images with controlled conditions). Each task-specific dataset contains 1000 training samples, with varying sample counts per class, and is evaluated using the original test set.
>
> It is important to note that with only 1,000 training images, different PEFT methods demonstrate their own strengths and weaknesses across diverse image domains. However, the crucial factor is whether a method can deliver impressive performance across all 19 tasks, showcasing robust and powerful transfer capabilities. Our HST method accomplishes exactly this, exhibiting outstanding performance across various domains: an average accuracy of 82.83% in Natural, 87.13% in Specialized, 64.45% in Structured, and an overall average accuracy of 76.0%, all of which currently stand as state-of-the-art performance benchmarks.
>
> Furthermore, we conducted experiments in general image classification using CIFAR-100 and four other FGVC datasets. Our HST method outperforms other PETL methods when using both ImageNet-22k and MAE pre-trained weights, further reinforcing the robust transfer capabilities inherent in HST. For detailed results, please refer to Table 6 in Appendix B.
>
> **Q5:  The performance on much large models**
>
> | Mask R-CNN 1x schedule (ViT-L) | Params(M) | APb  | APb(50) | APb(75) | APm  | APm(50) | APm(75) |
> |-|:-:|:-:|:-:|:-:|:-:|:-:|:-:|
> | Full-tuning| 337.3| 45.7 | 68.9| **49.4** | 41.5 | 65.6| **44.6**|
> | Head-tuning| 33.6| 31.6 | 56.4 | 32.0| 31.3 | 53.3 | 32.5|
> | LoRA-64| 39.84| 44.8 | 68.8 | 48.9| 41.2 | 65.3 | 44.0|
> | HST| 39.62| **45.7** | **69.1**| 49.2 | **41.6** | **65.6** | 44.4|
>
> We can observe that HST performs more satisfactorily when using larger models like ViT-L. There is a performance gap of 2.6 AP_b between HST and full finetune on the base model, while achieving comparative or even better performance on the large model. Additionally, the LoRA method also demonstrates further improvement in performance with larger models, although it still remains slightly lower than full finetune by approximately 1.0 AP_b.

---

### Author Response · Authors · 2023-11-19
**General Response to All Reviews (1/2)**

We would like to express our gratitude to all the reviewers for their constructive feedbacks. Firstly, we aim to further elaborate on the motivations and address some common issues.

**(1) Motivations and Sources of Inspiration**

Overall, in both prevailing CNN and Vision Transformer networks, there is a prevailing trend towards utilizing pyramid-style architectures as backbones. These architectures incorporate multi-scale feature enhancement to improve performance in various tasks.
However, many existing pre-training methods rely on plain ViT models, such as ImageNet-22k, CLIP, MAE, DINO, and SAM, etc. This raises a fundamental question: how can these pre-trained plain ViT models be efficiently adapted to dense prediction tasks?

Methods like ConvMAE and ITPN (HiViT), as mentioned by Reviewer ExGi, initially conceptualize the network as a multi-scale architecture before pre-training and subsequently apply it to downstream tasks. However, this approach inherently requires greater training resources.

Therefore, the idea is to equip the plain ViT with an "armor" that enables it to output multi-scale features similar to a pyramid model. ViT-Adapter precisely addresses this issue, and it is evident that a sophisticated "armor" (side network) performs better than a crude "armor" (upsample or downsample module).

However, these methods primarily focus on enhancing ViT's performance by employing full fine-tuning. In the current era of large-scale models, conducting full fine-tuning for each downstream task has become increasingly challenging and requires substantial storage space. Thus, the challenge persists in enhancing performance under parameter-efficient fine-tuning and our article is dedicated to addressing this challenge.

**(2) Detailed Architecture Specifications**

| | | |ViT| | | | |HSN| | | |
|-|:-:|:-:|:-:|:-:|:-:|:-:|:-:|:-:|:-:|:-:|:-:|
||embed_dim|depth|num_heads|mlp_ratio| \||embed_dims|num_heads|depths|MetaT_N|#Param|
| Base(Cls)|768|12|12|4|\||[32,48,64,72]|[2,4,8,12]|[3,3,3,3]|1|0.78M|
| Base(Det/Seg)|768|12|12|4|\||[64,128,256,384]|[2,4,8,12]|[3,3,3,3]|8|13.21M|
| Large(Cls)|1024|24|16|4|\||[32,48,64,72]|[2,4,8,12]|[6,6,6,6]|1|0.78M|
|Large(Det/Seg)|1024|24|16|4|\||[64,128,256,384]|[2,4,8,12]|[6,6,6,6]|8|19.86M|

Specifically, we set different dimensions and attention heads for each stage in HSN, gradually increasing as the layer depth. In classification experiments, HSN's dimensions [32, 48, 64, 72] are significantly smaller than ViT's (768), and we only use one MetaT, resulting in a substantial reduction in the number of training parameters. However, for dense prediction tasks, we choose slightly larger dimensions [64, 128, 256, 384] to ensure sufficient capacity for handling dense prediction tasks. Notably, neck modules like FPN also adopt dimensions of [64, 128, 256, 384], which sets them apart from other methods where neck modules maintain ViT's dimensions, thus requiring fewer training parameters.

**(3) Differences among HST, ST, LST, SAN and ViT-Adapter**

>**HST vs. Side Tuning (ST):**

ST is the most basic and straightforward method of side tuning. It involves learning a side model S(x) and combining it with a pre-trained base model B(x) in the last layer, without any interaction at the intermediate feature layers.

>**HST vs. Ladder Side-Tuning:**

The LST method was initially introduced in the field of NLP to address training efficiency issues. It involves freezing the pre-trained model and utilizing intermediate features as supplementary inputs to train a side network. This side network has the same architecture as the pre-trained model but with fewer layers. However, initializing the side network poses a challenge, which is overcome by extracting weight matrices from the larger model to initialize the smaller model's matrices. Differently, our HSN can be randomly initialized, providing additional flexibility.

When we attempted to directly apply the LST method to ViTs, it didn't bring satisfied results. We found that the side network struggled to discern and fit the intermediate features from the pre-trained model. Nevertheless, with the introduction of LN-Tuning, the intermediate features gradually adapted to the distribution of downstream tasks during the training process. This allowed the advantages of the side network to be fully demonstrated, which is one of reasons why we incorporated LN-Tuning into our HST.
Furthermore, it is notable that the side network in LST is still a plain Transformer, lacking the ability to efficiently handle dense prediction tasks and lacking spatial prior knowledge related to images.

In contrast, our HST is specifically designed to address these limitations. It serves as a versatile and parameter-efficient fine-tuning approach for pre-trained ViT models. By integrating refined modules such as MetaT, Meta-Globa/Fine-grained Injection, and a pyramid architecture, we significantly enhance the performance of ViT models in various downstream tasks.

---

> ### Author Response · Authors · 2023-11-19
> **General Response to All Reviews (2/2)**
>
> >**HST vs. Side Adapter Network for Open-Vocabulary Semantic Segmentation (SAN):**
>
> **(a) Motivation:**
>
> SAN enhances the capabilities of the existing VL pre-trained model (CLIP) by enabling open-vocabulary semantic segmentation. It achieves this by maintaining CLIP's open-vocabulary recognition by freezing pre-trained weight while using a side adapter network to propose masks for better alignment. Essentially, SAN is a specialized design for CLIP weights, offering a unique and efficient approach to open-vocabulary semantic segmentation.
>
> Conversely, HST is a new method in PETL, shares the same objective as LoRA and adapters. Its goal is to swiftly transfer pre-trained models to different downstream tasks, demonstrating superior performance.
>
> **(b) Architecture:**
>
> The side network in SAN resembles LST, both using plain Transformer models. However, SAN integrates trainable Query Tokens within the side network to produce mask proposals. These Query Tokens significantly differ from our Meta Tokens. The former, similar to methods like Maskformer[1], serve the purpose of generating mask proposals, while the latter, akin to prompt tuning, are fewer in number and offer global information to the side network. Furthermore, the output features from SAN's side network are injected back into the final layers of the pre-trained model, but this process is not present in LST or our HST.
>
> However, the side network in HST differs substantially from SAN and LST. It's a pyramid-style architecture generating multi-scale features, aiding dense prediction tasks. Additionally, we employ convolutional layers as embedding layers to introduce image-related spatial knowledge.Furthermore, our side network no longer incorporates self-attention; instead, it maximizes the use of feature information learned by MetaT for cross-attention. This approach not only circumvents the significant computational burden when dealing with high-resolution feature maps but also effectively injects pre-trained information into our HSN.
>
> [1] Cheng B, Misra I, Schwing A G, et al. Masked-attention mask transformer for universal image segmentation[C]//Proceedings of the IEEE/CVF CVPR. 2022: 1290-1299.
>
> >**HST vs. Vit-Adapter:**
>
> **(a) Motivation:**
>
> ViT-Adapter boosts dense prediction task performance by integrating an adapter network into the pre-trained ViT, enhancing its overall performance through joint full fine-tuning. Conversely, HST targets efficient parameter fine-tuning to achieve satisfactory results in various visual tasks like image classification and dense predictions.
>
> **(b)Architecture:**
>
> ViT-Adapter and HST share similarities in utilizing a plain ViT as their backbone and introducing a randomly initialized adapter/side network into the pre-trained ViT when transferring to downstream tasks, thereby decoupling the model structure between the upstream pre-training and downstream fine-tuning stages. However, they differ in several aspects:
> * Firstly, ViT-Adapter utilizes a spatial module to obtain three distinct features of different resolutions, concatting them into a unified entity via its adapter network for multi-scale outputs. In contrast, our HSN, a pyramid-style architecture, progressively captures features of various granularities throughout the network depth, producing multi-scale outputs at different stages.ViT-Adapter's interaction blocks impose a notable computational burden due to numerous input tokens. Conversely, our HSN mitigates this burden by utilizing a limited number of meta tokens, ensuring superior computational efficiency.
>
> * Secondly, ViT-Adapter incorporates bidirectional information flow by exchanging features between the ViT backbone and the adapter network, impacting both training and inference efficiency. In contrast, HST operates with a unidirectional information injection solely from the ViT backbone to the side network, resulting in enhanced training efficiency.
>
> * Thirdly, while ViT-Adapter incorporates sparse attention mechanisms, it requires significant training resources. Our attempts to train the adapter network while keeping ViT frozen within ViT-Adapter led to Out of Memory errors when using V100 GPUs, a problem that didn't occur with HST.

---

### Meta-Review · Area_Chair_aXyF · 2023-12-05

**Metareview:**

After discussion, major concerns about the motivation, implement details and fair comparison with existing works remain. After carefully reading the paper, the review comments, the AC deemed that the paper should undergo a major revision, thus is not ready for publication in the current form.

**Justification For Why Not Higher Score:**

N/A

**Justification For Why Not Lower Score:**

N/A

---

### Decision · Program_Chairs · 2024-01-16

Reject